# Fleas as Useful Tools for Science

**Pedro Marcos Linardi**

Departament of Parasitology, Institute of Biological Sciences, Federal University of Minas Gerais, Mailbox 486, Avenida Presidente Antônio Carlos, 6627, Belo Horizonte 31270-901, MG, Brazil; linardi@icb.ufmg.br

**Abstract:** Adult fleas are blood-feeding insects that exclusively infest mammals, acting as parasites and disease vectors. Although certain species exclusively inhabit nests, others are commonly found on the bodies of mammals. Immature stages develop in the soil, inside or near the nests of their respective hosts, making them susceptible to environmental alterations. On hosts, flea infestations are usually defined by abundance, prevalence, and diversity, varying according to host age, sex, size, behavior, habitat, and climate. However, in spite of their vast parasitological importance, fleas have only occasionally been used in applied research. This review focuses especially on the use of mammal fleas as tools or indicators in solving biological, epidemiological, ecological, and phylogenetic questions, and raises new perspectives for future studies.

**Keywords:** Siphonaptera; fleas; ectoparasites; hematophagous insects

## 1. Introduction

In general, insects are classified as useful or harmful according to their relationships with humans; the latter being those that draw the most attention regarding damage to agriculture and stored products and especially for disease transmission. This last category includes fleas (Siphonaptera), which are well known for their dual action as infesting agents and disease vectors. Adult fleas of both sexes are obligatory hematophagous solenophages of warm-blooded vertebrates, while larvae develop on the feces or any proteinaceous matter. Flea metamorphosis is complete and, except for the genus *Tunga* [1], there are three larval stages. As ectoparasites, fleas provoke irritations to host skin and can produce allergic reactions and secondary infections, such as with tungiasis. On the other hand, when acting as vectors, they transmit flea-borne diseases, the most common of which are presented in Table 1, with certain particularities of their pathogens [2].

Despite their vast parasitological importance and the numerous works on their taxonomy, chemical control, and ecology related to richness, abundance, and prevalence, some actions of fleas can be used in practical scientific applications. This review focuses on the use of fleas as tools or indicators in solving some biological questions, especially during mammal infestation, and raises new perspectives for future studies.

**Table 1.** Flea-borne diseases affecting humans and other animals. Source: Linardi [2] (reproduced with the permission of Springer Nature).

| Disease | Pathogen | Localization of Pathogen in the Flea | Reproduction of Pathogen in Flea | Reservoirs | Flea Species Vectors | Method of Transmission | Pathogenic Effect on the Flea |
|---|---|---|---|---|---|---|---|
| Myxomatosis | Myxoma virus | Digestive tract | − | Rabbits: *Oryctolagus cuniculus, Sylvilagus* | *Spilopsyllus cuniculi* | Mechanical inoculation | + |
| Murine typhus | *Rickettsia typhi* | Digestive tract | + | Commensal rats | *Xenopsylla cheopis, Ctenocephalides f. felis* | Feces and crushing | − |
| Flea-borne spotted fever | *Rickettsia felis* | Digestive tract | + | Cats | *Ctenocephalides f. felis* | Crushing, bite | − |
| Cat scratch disease | *Bartonella henselae* | Digestive tract | + | Cats and wild rodents | *Cenocephalides f. felis, C. canis, Polygenis gwyni* | feces | − |
| Salmonellosis | *Salmonella enteritidis, S. typhimurium* | Digestive tract | + | Rodents and man | *X. cheopis, Nosopsyllus fasciatus* | Mechanical inoculation and contamination | + |
| Tularemia | *Francisella tularensis* | Digestive tract | − | Rodents and lagomorphs | *Diamanus montanus, Cediopsylla simplex, Neopsylla setosa, Ctenophthalmus assimilis* | Mechanical inoculation and contamination | − |
| Bubonic plague | *Yersinia pestis* | Digestive tract | + | Wild and commensal rodents | *X. cheopis, X. brasiliensis, X. astia, N. fasciatus, Polygenis* spp. | Bite | + |
| Murine trypansomatids | *Trypanosoma lewisi* | Digestive tract | + | Synantrhropic rodents | *X. cheopis, N. fasciatus* | Feces | − |
| Dilepidiasis | *Acanthocheilonema (=Dipylidium caninum)* | Body cavity | − | Dogs | *C. felis felis, C. canis, P. irritans* | Ingestion, crushing | + |
| Hymenolepiasis | *Hymenolepis diminuta, Rodentolepis (=Hymenolepis) nana* | Body cavity | − | Synanthropic rodents | *X. cheopis, N. fasciatus, Leptosylla segnis, P. irritans, Cteno- cephalides* spp. | Ingestion, crushing | + |
| Canine filariasis | *Dipetalonema reconditum* | Body cavity | − | Dogs | *C. felis felis, C. canis* | Active penetration after biting | + |

## 2. Epidemiology

Due to its paramount importance in the transmission of bubonic plague and murine typhus, *Xenopsylla cheopis* (Rothschild, 1903) is the most cited flea species in studies worldwide. Despite its disastrous action, this species can be used in plague surveillance to signal dangerous situations by means of pulicidian indices. Although they vary seasonally and are not indicative of nest populations, these indices are customarily calculated to demonstrate seasonal trends. One such index is the Flea Index, which is calculated as the total number of fleas recovered from all trapped animals divided by the number of trapped animals. Another is the Specific Flea Index (SFI), which is calculated by dividing the number of individuals of a particular flea species, especially *X. cheopis*, by the total number of host animals. An SFI for *X. cheopis* of over 1 is indicative of a dangerous situation regarding bubonic plague [3–5].

As a pernicious agent, fleas have been used as instruments of war. During the 1346 siege of Caffa (Ukraine), the Tartars used the bubonic plague as a biological weapon,

hurling corpses via catapults against Christians in the hope that the intolerable stench would kill everyone [6]. During World War II, Japan used *X. cheopis* infected with *Yersinia pestis* when bombing China [7,8]. The sand flea, *Tunga penetrans* (L., 1758), is another species with potential as a biological weapon, given its natural habitat is primarily the sandy and warm soils of deserts and beaches [9], where larvae are found at a depth of 2–5 cm in the sand [10]. These authors express the need for further studies of larval behavior and their vertical migration in sand. Due to the infestation it causes, an anecdotal application was also proposed, suggesting the species might be used in the defense of Brazilian territory in the case of a foreign invasion, due to the devastation it would cause to enemy troops [11].

Infection by *Trypanosoma lewisi* does not have any direct pathogenic effect on *X. cheopis* (Table 1); however, this trypanosomiasis may indirectly cause some harm to rodents. Under experimental conditions, *T. lewisi* was found to increase the multiplication of *Toxoplasma gondii* in white rats [12,13]. This finding deserves greater attention, since (a) rats infected with *T. gondii* are considered important in the epidemiology of toxoplasmosis because they can serve as reservoirs of infection for pigs, dogs, and cats; and (b) *T. gondii* infection may enhance the likelihood of infected rats being predated by cats. Thus, if increasing the level of *X. cheopis* infestation of rats favors *T. lewisi* transmission, which in turn promotes the spread of *T. gondii*, flea control might result in the control of rat toxoplasmosis [14].

The most classic use of fleas for biological control is the use of the European rabbit flea. This species exclusively infests wild rabbits (*Oryctolagus cuniculus*), carrying with them the virus that causes myxomatosis. Wild rabbits were introduced to Australia in the mid-to-late 19th century and quickly became invasive pests, heavily damaging newly developed livestock industries and agriculture, as well as soils and native flora, even eliminating native fauna. *Spylopsyllus cuniculi* (Dale, 1878) was released among Australian wild rabbits and caused a high mortality, especially among young hosts [15–17]. The success of this operation was evidenced by pasture regeneration, a sheep population increase, and economic strengthening.

Allantonematid worms might also have use in biological control because they can injure parasitized hosts by sterilization and atrophy of genitalia and can even be lethal [18], as well as cause the feminization of males and masculinization of females of certain flea species [19]. These nematodes were found in body cavities of the Neotropical flea *Polygenis (Polygenis) tripus* (Jordan, 1933) with a prevalence of 11.2% [20]. Considering that species of *Polygenis* maintain plague among wild rodents, and that *P. tripus* is the principal species of the genus in Brazil, the authors highlighted the possibility of its use for biological control and consequent prophylaxis for the plague.

Of the various endosymbionts identified in 60 species of fleas developing mutualistic, commensalistic, or parasitic actions [21], little of significance is known about their biology, prevalence, and degree of pathogenicity for natural flea populations. Molecular data have increased the possibility of finding other endosymbionts, such as *Wolbachia* spp., which was observed for the first time among species of Siphonaptera, including *Ctenocephalides canis* (Curtis, 1826), *C. felis felis* (Bouché, 1835), *Echidnophaga gallinacea* (Westwood, 1875), *Orchopeas howardii* (Baker, 1895), *Polygenis gwyni* (C. Fox, 1914), and *Pulex irritans* (L., 1758) [22]. *Wolbachia* spp. subsequently found in (i) 20 more species of the Rhopalopsyllidae, Stephanocircidae, Pulicidae, Ceratophyllidae, Ctenophthalmidae, Ischnopsyllidae, Leptopsyllidae, and Malacopsyllidae families from sylvatic populations throughout the Nearctic and Neotropical regions [23]; (ii) *Tunga trimamillata* (Pampiglione et al., 2002) from Ecuador [24]; (iii) *T. penetrans* from Brazil [25]; and (iv) *Ctenocephalides* spp. and *P. irritans* from Turkey [26].

Other symbionts, namely *Nolleria pulicis* (Chitridiopsidae) and a gregarine, *Steinina* sp. (Actinocephalidae), were found in *C. felis felis* collected from dogs in Belo Horizonte, Minas Gerais, Brazil [27]. The gregarine was observed in different stages, with 16.8% of the infected fleas carrying more than 20 gametocysts obstructing the midgut (Figure 1), in spite of the acceleration of the biological cycle of the flea [28]. *Ctenocephalides felis felis* is currently the most studied flea species due to efforts to control pet infestation, which raises

the question of the extent to which some endosymbionts might be pathogenic for fleas and thus be used in biological control.

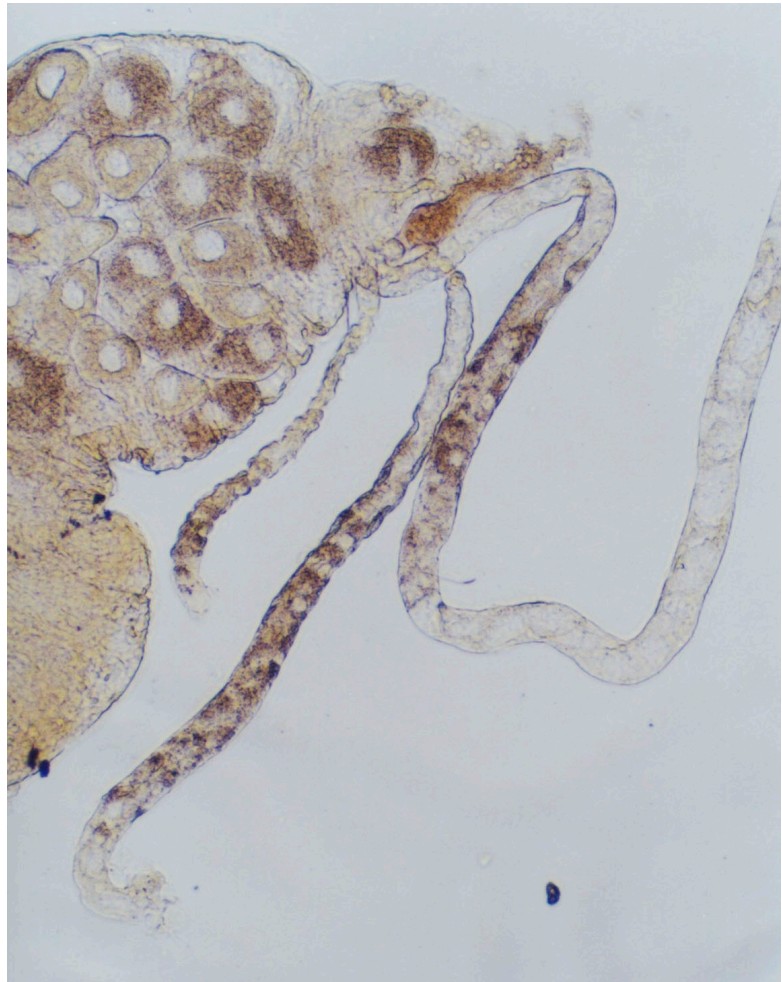

**Figure 1.** Trophozoites of *Steinina* sp. in the midgut of *C. felis felis*.

## 3. Taxonomy

Although the name of the order remained in dispute for several years [29], the common name for the group "fleas" has been known since ancient times, given the biblical citations:

"After whom has the king of Israel come out? After whom do you pursue? After a dead dog! After a flea!" (1 Samuel 24:14) and "For the king of Israel is come out to seek a flea" (1 Samuel 26:20). There are about 2600 currently known species and/or subspecies of fleas, which are included in 238 genera and 15 families [30]. Most studies have been on the alpha taxonomic level, including identifications, new hosts and geographical records, local studies, and descriptions of new species and alternative sexes. Although siphonapterists work in several areas, the best known are taxonomists when describing species or even assisting other researchers in identifying specimens. Two authors are responsible for describing almost 20% of the world's flea species and subspecies: Jordan has described 258, Rothschild has described 232; and the two authors have jointly described another 184.

The world's leading flea taxonomists are listed in Table 2, along with number of species they described as a sole author and as a co-author, their period of activity, and the number of species described in their honor (based on data obtained until 2022) [31], not considering synonyms.

**Table 2.** Some authors who have described flea species and subspecies.

| | Number of Species and Subspecies | | | | |
| | Described | | | Named in Honor | Period |
| | Author | Co-Author | Total | | |
|---|---|---|---|---|---|
| Rothschild | 192 | - | 192 | 5 | 1897/1923 |
| Jordan | 240 | 187 [1] | 427 | 6 | 1906/1958 |
| Wagner | 112 | 16 [2] | 128 | 6 | 1893/1939 |
| Baker | 35 | - | 35 | - | 1895/1905 |
| Ioff | 98 | 33 | 131 | 7 | 1926/1953 |
| DeMeillon | 22 | 13 | 35 | 1 | 1930/1960 |
| Liu | 4 | 60 [3] | 64 | 8 | 1939/2006 |
| Hubbard | 38 | - | 38 | 2 | 1940/1967 |
| Traub | 86 | 28 | 114 | 14 | 1944/1977 |
| Holland | 55 | 4 | 59 | 4 | 1949/1979 |
| Peus | 55 | - | 55 | - | 1950/1978 |
| Smit | 167 | 16 | 183 | 14 | 1950/1987 |
| Li | 15 | 28 [3] | 43 | 7 | 1957/1999 |
| Wu | | 34 [3] | 34 | 2 | 1960/2007 |
| Lewis | 41 | 15 | 56 | 5 | 1962/2001 |
| Beaucournu | 27 | 141 | 168 | 2 | 1962 |
| Mardon | 20 | 18 | 38 | 1 | 1971/1986 |
| Hastriter | 63 | 13 | 76 | 1 | 1975 |
| Xie | 1 | 29 [3] | 30 | 2 | 1960/2007 |

[1] All descriptions made with Rothschild. [2] Ten descriptions with Ioff. [3] Only as a first author.

Fleas, lice, ticks, and mites are now often studied simultaneously, particularly as part of surveys or of a collection of mammals. When multiple infestations among ectoparasites are found, they are an important way to confirm the identification of hosts. Molecular analyses of DNA are also being used for both identifying ectoparasites and determining their phylogenetic relationships [32], although there are few studies of fleas that have analyzed genetic diversity using preserved specimens [33–35]. For flea samples that are considered to have been inadequately preserved in 70% ethanol for long periods, Multiple Displacement Amplification (MDA) of siphonapterid DNA holds promise as a valuable tool for studies concerning taxonomy, phylogeny, and epidemiology [36]. The combination of nuclear and mitochondrial markers has also been proposed as useful to identify the two sexes of species of the larger genus *Ctenophthalmus*, considering that only males can be identified using morphological data [37].

More consistent studies for the identification of species are currently being carried out, associating morphological and molecular data with integrative taxonomy. The advent of artificial intelligence (AI) is changing the way insects are identified, because traditional identification is often labor-intensive. Machine learning algorithms, a subset of AI, can be trained to recognize and classify insects based on images. Although such technology has yet to be used in flea taxonomy, it likely will be soon.

## 4. Archaeology

Human fleas have been recovered from Tell el-Amarna, Egypt, dating from about 3550 years ago, and also in large numbers from Norse Greenland, medieval Dublin, Anglo-Scandinavian York, 18th-century London, and Dutch sites from between the 11th and 14th centuries [38]. Hundreds of specimens of *P. irritans* were recovered from mummified dogs in southern Peru dating from around 900 A.D.; thus, the presence of the species in the Americas before European colonization has been confirmed [39].

Unlike lice, whose adults and nymphs do not survive off hosts and the nits remain attached to host hair even after death, fleas abandon their hosts when disturbed, stressed, or killed within 24 h after death [40], or even before, and in 2–4 h after snap-trapping [41]. More than 1200 specimens of *Pulex* sp. were encountered on the fur of mummies of *Cavia*

*porcellus* and *Canis familiaris* in an archaeological site in southern Peru. The discovery of fleas indicates that these hosts were buried immediately or not long after death, because these ectoparasites would never invade a dead or moribund animal [41].

Although the subject of this review is mammalian fleas, eiderdown production observed in Icelandic archaeological sites was related to the presence of bird fleas, mainly *Ceratophyllus garei* (Rothschild, 1902), representing the first attempt at identifying the entomological signature of this activity [42].

## 5. Biology

For most flea species, females predominate over males when caught on hosts. In general, female fleas live longer than males [40].

It is important to emphasize that although the sex ratio at emergence is equal, this does not always occur. Of 207 flea collections belonging to 108 species, 20% had no significant imbalance, 2% were predominately male, and 78% were predominately female. Samples obtained from both bodies and nests showed similar results [43]. Data for 57 samples of Neotropical flea species belonging to eight families and totaling 50,633 specimens collected from Argentina, Bolivia, Brazil, Chile, Panama, Peru, and Venezuela are shown in Table 3. Of these, 47 samples were female-biased, five had a sex ratio close to unity, and five were male-biased (Table 3), in accordance with other authors [43,44]. The following female/male sex ratios were observed for flea families: Ceratophyllidae (1.22), Ctenophthalmidae (1.19), Ischnopsyllidae (2.39), Leptopsyllidae (1.93), Pulicidae (0.93), Rhopalopsyllidae (1.36), Stephanocircidae (1.79), and Tungidae (1.40).

**Table 3.** Sex ratio of some flea species from Neotropical mammals.

| Species | (N) | ♀/♂ Ratio | Reference |
|---|---|---|---|
| *Adoratopsylla (A.) a. antiquorum* (Rothschild, 1904) | 595 | 1.14 | USP [1] |
| *Adoratopsylla (A.) a. antiquorum* | 41 | 1.15 | UFMG [2] |
| *Adoratopsylla (A.) dilecta* Jordan, 1938 | 66 | 1.10 | [45] |
| *Adoratopsylla (A.) discreta* (Jordan, 1926) | 72 | 1.25 | [45] |
| *Adoratopsylla (Tritopsylla) i. intermedia* (Wagner, 1901) | 592 | 1.30 | [45] |
| *Adoratopsylla (T.) i. intermedia* | 80 | 0.95 | UFMG |
| *Adoratopsylla (Tritopsylla) i. copha* (Jordan, 1926) | 761 | 1.08 | [46] |
| *Cleopsylla monticola* Smit, 1953 | 142 | 1.58 | [45] |
| *Craneopsylla m. minerva* (Rothschild, 1903) | 71 | 2.38 | UFMG |
| *Ctenocephalides f. felis* | 1088 | 2.14 | [46] |
| *Ctenocephalides f. felis* | 268 | 3.18 | UFMG |
| *Gephyropsylla k. klagesi* (Rothschild, 1904) | 2313 | 1.39 | [46] |
| *Gephyropsylla k. klagesi* | 618 | 1.21 | [45] |
| *Gephyropsylla k. samuelis* (Jordan and Rothschild, 1923) | 754 | 1.13 | [45] |
| *Hechtiella nitidus* (Johnson, 1957) | 152 | 1.14 | UFMG |
| *Hormopsylla fosteri* (Rothschild, 1903) | 123 | 2.72 | [47] |
| *Jellisonia johnsonae* (Tipton and Méndez, 1961) | 79 | 1.39 | [46] |
| *Juxtapulex echidnophagoides* (Wagner, 1933) | 599 | 1.19 | [46] |
| *Kohlsia traubi* Tipton and Méndez, 1971 | 365 | 0.99 | [46] |
| *Leptopsylla segnis* (Schönherr, 1811) | 4532 | 1.95 | [48] |
| *Leptopsylla segnis* | 304 | 1.78 | [49] |
| *Leptosylla segnis* | 16,080 | 1.84 | [50] |
| *Leptopsylla segnis* | 56 | 1.24 | [45] |
| *Neotyphloceras crassispina hemisus* Jordan, 1936 | 49 | 2.06 | [51] |
| *Neotyphloceras rosenbergi* (Rothschild, 1904) | 197 | 1.28 | [45] |
| *Nosopsyllus fasciatus* (Bosc, 1800) | 156 | 2.00 | [48] |
| *Pleochaetis altmani* (Tipton and Méndez, 1961) | 116 | 1.00 | [46] |
| *Pleochaetis d. dolens* (Jordan and Rothschild, 1914) | 479 | 1.46 | [46] |
| *Pleochaetis d. quitanus* (Jordan, 1931) | 272 | 1.22 | [45] |
| *Pleochaetis smiti* Johnson, 1954 | 411 | 1.02 | [45] |

**Table 3.** *Cont.*

| Species | (N) | ♀/♂ Ratio | Reference |
|---|---|---|---|
| *Plocopsylla pallas* (Rothschild, 1914) | 69 | 1.46 | [51] |
| *Plocopsylla ulisses* Hopkins, 1951 | 58 | 1.52 | [45] |
| *Polygenis (Polygenis) b. bohlsi* (Wagner, 1901) | 117 | 1.29 | [45] |
| *Polygenis (P.) b. bohlsi* | 126 | 1.63 | UFMG |
| *Polygenis (Polygenis) b. jordani* (Lima, 1937) | 341 | 1.48 | USP |
| *Polygenis (Polygenis) dunni* (Jordan and Rothschild, 1922) | 231 | 1.65 | [45] |
| *Polygenis (Polygenis) rimatus* (Jordan, 1932 | 208 | 1.24 | UFMG/MLP [3] |
| *Polygenis (Polygenis) roberti beebei* (Fox, 1947) | 118 | 1.80 | [46] |
| *Polygenis (P.) roberti beebei* | 266 | 1.33 | [45] |
| *Polygenis (Polygenis) tripus* (Jordan, 1933) | 578 | 1.51 | UFMG |
| *Polygenis (Neopolygenis) pradoi* (Wagner, 1937) | 88 | 1.93 | USP |
| *Ptilopsylla dunni* Kohls, 1942 | 441 | 1.95 | [46] |
| *Pulex irritans* | 679 | 0.91 | UFMG |
| *Pulex simulans* Baker, 1895 | 84 | 1.70 | [46] |
| *Rhopalopsyllus australis australis* (Rothschild, 1904) | 225 | 1.74 | [45] |
| *Rhopalopsyllus australis tupinus* (Jordan and Rothschild, 1923) | 226 | 1.35 | [46] |
| *Rhopalopsyllus lugubris cryptotecnes* (Enderlein, 1912) | 125 | 1.25 | [46] |
| *Rhopalopsyllus saevus* (Jordan and Rothschild, 1923) | 181 | 1.35 | [46] |
| *Sphinctopsylla tolmera* (Jordan, 1931) | 112 | 2.11 | [45] |
| *Sternopsylla d. distincta* (Rothschild, 1903) | 91 | 7.27 | UFMG] |
| *Tiamastus palpalis* (Rothschild, 1911) | 58 | 1.41 | [52] |
| *Tunga penetrans* | 215 | 1.86 | [53] |
| *Xenopsylla brasiliensis* (Baker, 1904) | 6086 | 0.67 | [48] |
| *Xenopsylla brasiliensis* | 1758 | 0.83 | [49] |
| *Xenopsylla cheopis* | 3624 | 0.95 | [48] |
| *Xenopsylla cheopis* | 1823 | 1.19 | [49] |
| *Xenopsylla cheopis* | 1274 | 0.89 | UFMG |

[1] Universidade de São Paulo, Brazil [2] Universidade Federal de Minas Gerais, Brazil [3] Museo de La Plata, Argentina.

It is interesting to note that the highest female-biased sex ratios are observed in fleas with two or more combs (ctenidia)—*C. felis felis*, *L. segnis*, bat fleas, and helmet fleas—thus agreeing in part with another study [54] which suggested that male fleas have smaller combs than females and thus would be less successful at host attachment. On the other hand, how often could it be expected that for species exhibiting a high imbalance in favor of females that they would emerge earlier and live longer than males? Females also predominate in samples from nests, and the reasons for imbalance on both the host's body and in nests might be due to sampling methods, unequal longevity, activity, age of hosts, ability to face adverse situations, and morphological differences [43,44].

The stimuli responsible for fleas finding their hosts are mainly olfactory, as for *X. cheopis* [55], preponderantly urine-based, as for *S. cuniculi* [56], or visual and thermal, as for *C. felis felis* [57]. Light and carbon dioxide stimulate flea locomotion and displacement to the emitting source. Consequently, positive phototaxis and chemotaxis form the basis for the use of traps in the control of certain flea species [58].

Infestations vary according to host age, sex, size, behavior, mobility, habitat, and climate [40], all of which are factors that should be mentioned in publications for possible practical applications concerning parasitism. For example, male-biased infestation, as observed on rats, may exist because male rats have larger home ranges, are generally larger than females, and exhibit territorial behavior. Other sex and age preferences for hosts result from grooming, because males are more efficient groomers than females and adults groom more than young individuals. On the other hand, a preference for female hosts might be related to hormonal cycles, as observed between *S. cuniculi* and *O. cuniculus* and between *Cediopsylla simplex* (Baker, 1895) and *Sylvilagus* spp. On this subject, M. Rothschild [59] raises the following question: *In the old literature it was repeatedly stated that women are attacked more frequently by fleas than men are. This has been generally attributed to the more*

*delicate skin and more sensitive nature of the fair sex. In old books it is always women who are pictured wearing the latest flea trap. Perhaps this faulty reasoning and the truth of the matter is that the human flea (Pulex irritans) also responds to the attraction of the ovarian hormones. This is food for reflection. . .* In such traps (Figure 2), a perforated wood or ivory cylinder, possibly holding blood-soaked cloth, was supposed to attract and capture fleas. However, what would have been the origin of the blood placed inside the traps? Menstrual blood?

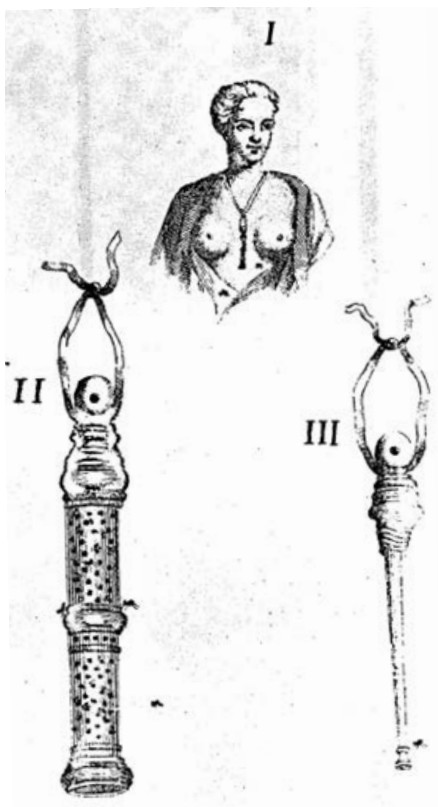

**Figure 2.** Flea trap depicted in a German book from 1739 and reproduced by Rothschild [59].

## 6. Ecology

Fleas have 60 million years of evolutionary history, even being found on prehistoric mammals [60]. Flea hosts are endothermic animals, of which approximately 94% are mammals. The following mammalian orders have known instances of infestation by fleas [40]: Rodentia (74%), Insectivora (8%) Marsupialia (5%), Chiroptera (5%), Lagomorpha (3%), and Carnivora (3%). Less than 1% of records of infestation are found in Monotremata, Cingulata, Pilosa, Pholidota, Hyracoidea, Artiodactyla, Perissodactyla, and Proboscidea. Among species of primates, only humans are regarded as a usual host. Parasitism on birds is secondary and must have originated from fleas parasitizing mammals, since the majority of bird fleas are associated with various groups of sea birds [61]. Another argument is that among the 14 families of the order, 10 do not contain any genus or species that infests birds; avian parasitism by fleas would be more ecological than phyletic [62]. An early association of fleas with mammals was also revealed based on the molecular phylogeny of Siphonaptera [35]. Among mammals, Rodentia is the most important order regarding fleas because it (i) contains the greatest number of parasitized species; (ii) is widely distributed geographically; (iii) contains species that occupy various niches in different ecotopes; and (iv) contains species that function as reservoirs of infections transmitted by fleas (e.g., plague, murine typhus, tularemia).

Flea hosts can be classified by their relationships as natural or casual associations [40, 44,61,63]. Thus, hosts can be considered as true, primary, accidental, or secondary for a given species of flea. True hosts, also called normal, essential, or primary hosts, are those

that provide favorable conditions under which a flea species can reproduce indefinitely. Evolutionary primitive or ancient hosts tend to be infested more by primitive fleas, while more evolved hosts tend to be associated with more recent or specialized ectoparasites [64]. Host specificity is the rule rather than the exception with fleas [64]. This is doubly enforced because larvae are essentially free-living insects with different nutritional and environmental requirements than adults. Approximately 600 flea species are specific, each infesting a single host species, being known through a single record [65]. On the one hand, specific associations, characterized by host exclusivity, constitute an auxiliary means for the identification of respective hosts, and may support mastozoologists. On the other hand, the eclectic nature of certain species, due to host diversity and polyhematophagism, is an important parameter in the study of epidemiological issues, given the exchange of hosts and disease transmission.

Fleas are distributed from the Arctic to Antarctica, with temperate regions having greater flea species richness, both latitudinally and longitudinally. This distribution is due to the greater preference of rodents for colder climates [63]. Geographic distributions of flea species are probably related to continental drift and plate tectonics, with subsequent dispersal and redistribution of host taxa [64]. Except for a few introduced forms, and considering Europe and Asia as a whole (i.e., Eurasia), no genus of mammal flea is found on four or more continents; only two occur on three continents, among them being *Tunga*. The Palearctic is the geographic region with the most diverse siphonapterofauna, representing 38% of the total number of known species, with the remainder being distributed similarly among other geographic regions. The number of flea species in the Neotropics (289) is similar to that in the Nearctic (299) and the Afrotropics (275), but less than that in the Palearctic (892). On the other hand, the percentage of endemic genera reaches 61% in the Afrotropics, followed by Australia (58%), the Neotropics (56%), the Palearctic (45%), the Orient (42%), and the Nearctic (37%) [44]. The diversity of fleas in some Latin American countries is presented in Table 4.

**Table 4.** Richness of mammals and fleas in some countries of Latin America.

| Countries | Extension km$^2$ | Mammals | | Fleas | |
|---|---|---|---|---|---|
| | | N | Reference | N | Reference |
| Argentina | 2,766,889 | 432 | [66] | 127 | [67] |
| Bolivia | 1,098,875 | 406 | [68] | 30 | [69] |
| Brazil | 8,511,966 | 775 | [70] | 65 | Linardi, unpublished |
| Chile | 751,625 | 163 | [71] | 112 | [72] |
| Colombia | 1,138,915 | 543 | [73] | 44 | [74] |
| Ecuador | 461,475 | 489 | [75] | 41 | [76] |
| Mexico | 1,972,545 | 544 | [77] | 172 | [78] |
| Panama | 78,515 | 251 | [79] | 37 | [46] |
| Peru | 1,285,215 | 573 | [80] | 77 | [81] |
| Venezuela | 912,045 | 390 | [82] | 54 | [45] |

It is surprising that Brazil, despite its vast area, expressive mammalian fauna, and the fact it is a hotspot of global biodiversity, has a low flea diversity, especially when compared to that of other countries with smaller geographical extensions and fewer known mammal species. Although the reasons for flea paucity in Brazil are deserving of further investigation, it is important to consider the subregions in which the countries listed in Table 4 are included. The Neotropical region includes the Brazilian and Patagonian subregions, with the former being subdivided into the Middle American and South American Provinces [83], also considered by Alfred Russel Wallace as Brazilian, Chilean, Mexican, and Antillean subregions (Figure 3). Contrary to Brazil, Colombia, and Venezuela, which are entirely within the Brazilian subregion, the countries of Argentina, Bolivia, Chile, Ecuador, and Peru are in two subregions, and Panama is situated in the Middle American Province. Countries that extend into two subregions, or are included only in the Patagonian sub-

region, have a greater flea diversity than those situated entirely within in the Brazilian subregion. The greater flea richness in Mexico is due to its inclusion in both the Neotropical and Nearctic regions.

Flea infestations on hosts are usually defined by their mean abundance and prevalence, and most ecological studies deal with these two parameters of collections or surveys in limited regions. From an epidemiological point of view, the mean abundance and the prevalence have different meanings. The mean abundance can be employed as an indicator of host health status, both specific and individual. Thus, a high abundance might be related to an inability of the host to oppose the action of the parasite by means of its immune system and/or its behavior (e.g., grooming) [84]. In this respect, the effects of host age and sex on parasite abundance require further investigation because the defensive capacity of mammals may well increase or decrease over time and ectoparasite grooming might be more prevalent in one of the sexes. Multiple infestations of host species by ectoparasites of different taxonomic groups may also influence the flea mean abundance, as these infestations are mediated by several types of ecological associations (intra- and inter-specific competition, predation, mutualism, etc.). Furthermore, an increase in the flea mean abundance might reflect an increasing mortality within the host population following infection by a pathogen, as with rodents and the bubonic plague.

Prevalence is related to the propagation of ectoparasites on their respective hosts. Thus, a high prevalence might be the result of micro-environmental overlap among hosts that, when associated with environmental factors, would favor the development of immature stages. In this context, prevalence would be related to spatial factors, including host territoriality and dispersion. Given the vectorial capacity of fleas, prevalence has the potential to measure the dissemination of pathogens. When abundance and/or prevalence are related to the certain factors, especially in studies with fleas and mites parasitizing the same hosts species, some practical results can be observed, such as (i) flea species richness increases with latitude of the center of host geographical range [85]; (ii) the abundance of the host's body occupied by fleas can be reliably used as an indicator of the entire flea population size, because indices of fleas on host bodies and of fleas in host burrows are positively correlated [86]; (iii) the relationship between the number of flea species and the number of flea genera per host tends to decrease with an increasing local mean annual temperature [87]; (iv) both in fleas and mites, the mean abundance predicts prevalence, and the prevalence of a flea species increases with an increase in its mean abundance within and across host species [88,89]; (v) the sex ratio of fleas collected from an individual host does not differ significantly from the sex ratio of the entire flea population [90]; and (vi) the flea niche breadth, measured in terms of both their spatial (geographic range size) and biological (host specificity) components, increases at higher latitudes in accordance with hypotheses about latitudinal gradients [91]. Still in relation to environmental factors, a study involving three hosts—*Monodelphis domestica*, *Necromys lasiurus*, and *Oligoryzomys eliurus*—and different localities across Brazil found that the mean flea abundance significantly increased with increasing mean annual air temperatures and proximity to the equator. Abundance also decreased with altitude for both *M. domestica* and *N. lasiurus* [92]. More recently, another study including 103 flea species showed that, for only males, the body size increases with latitude, in accordance with Bergmann's Rule [93]. In contrast, no relationship was observed between latitude and flea or mite abundance for samples collected from small mammals in the Palaearctic [94].

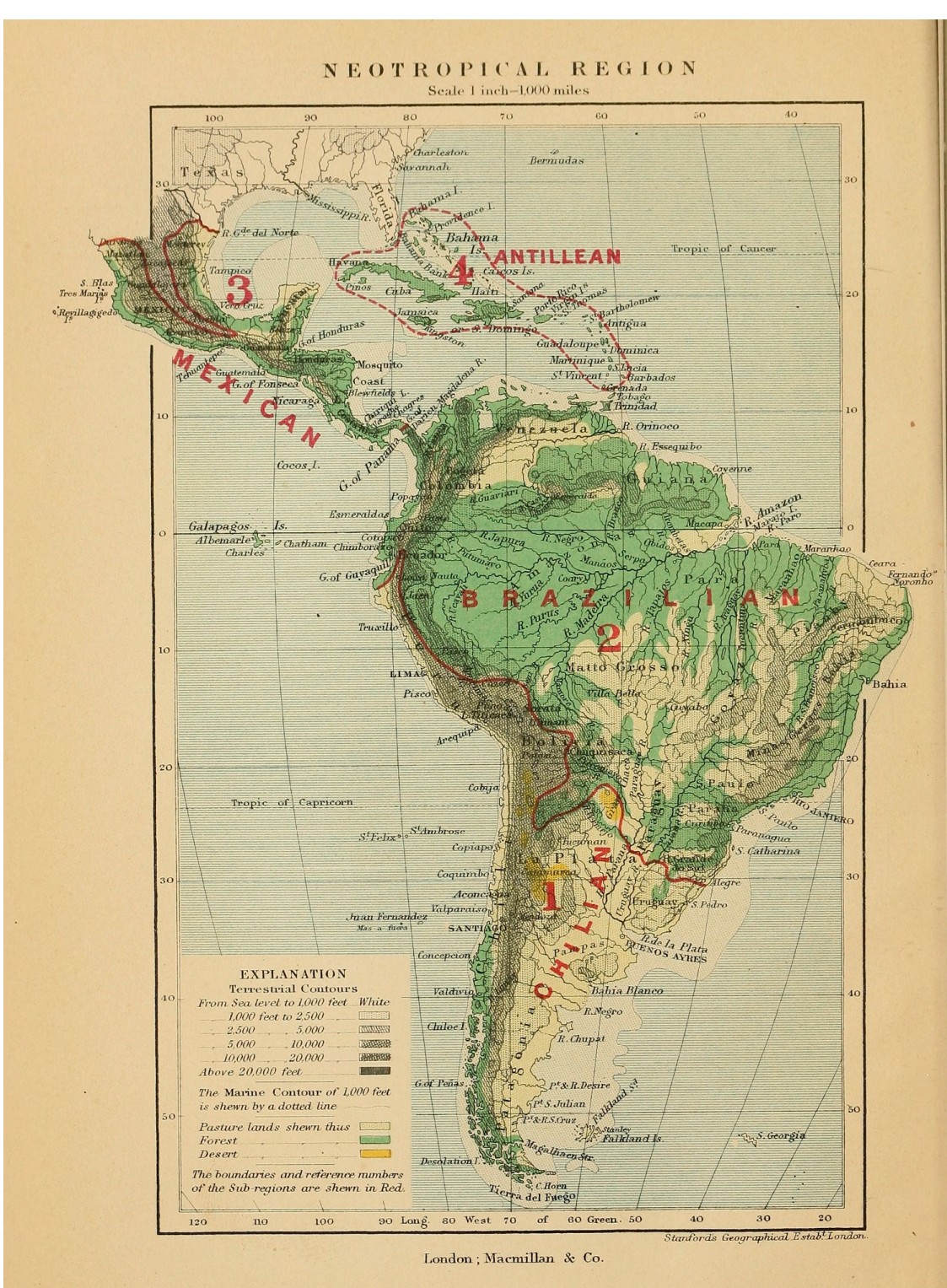

**Figure 3.** Neotropical region map including subregions according to Alfred Russel Wallace, *The Geographical Distribution of Animals* (S718: 1876). Source: web, licensable Figure: 1. Chilian subregion; 2. Brazilian subregion; 3. Mexican subregion; 4. Antillean subregion.

An ecological evaluation of habitats on Maracá Island, Roraima, Brazil, employed a comparative study of mammalian ectoparasite fauna using ectoparasites as characters. Maracá is a huge riverine island, located approximately 110 km northwest of Boa Vista and situated near the junction of the Amazonian forest and the dry savanna. The interchange of ectoparasites between hosts in two principal habitats (Amazonian forest and savanna) was

compared to a disharmonic relationship between predator and prey, in that the ectoparasites of one host can be acquired by the other through predatory action. Thus, in a habitat that expands to the detriment of another, the tendency would be towards a greater acquisition of ectoparasite species in the direction of expansion. The study showed a greater impact of forests on savannas than vice versa, indicating that the savanna is currently suffering retraction (Figure 4) [95].

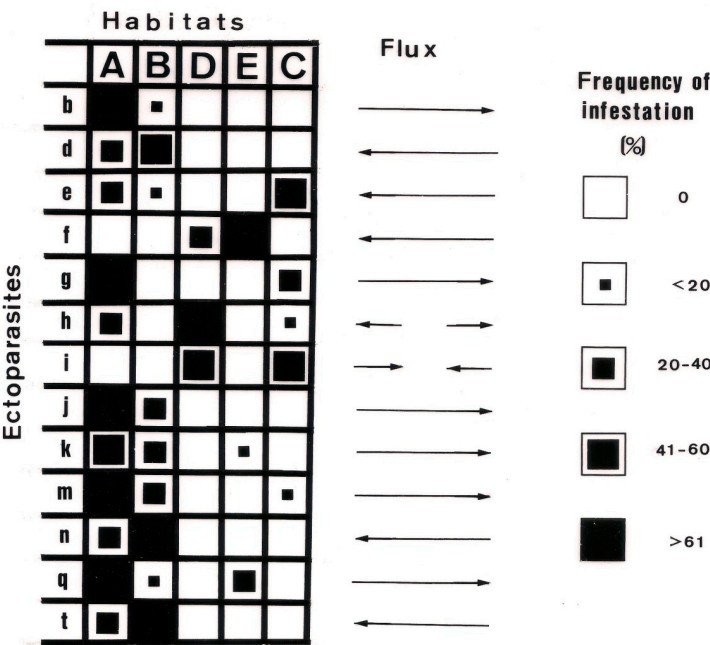

**Figure 4.** Interchange of ectoparasites between habitats on the Maracá Island, Roraima, Brazil, obtained from the frequencies of infestation on hosts in the respective habitats. A: dry savanna; B: forest; C: seasonally flooded savanna; D: ecological station; E: water tank area; b–t: species of mites, lice, and fleas infesting mammals. Source: Linardi and Botelho [95] (reproduced with permission from *Research Trends*).

Modern analytical methods of community ecology have also been applied to reveal patterns in historical biogeographies, such as in a comparative study of the phylogenetic structure of flea assemblages collected from small mammals on opposite sides of the Bering Land Bridge [96].

The application of a new ecological concept, called dark diversity, has led to a better understanding of the factors affecting species richness and the composition of communities [97,98]. Dark diversity is defined as the set of species that are absent from a study site but present in the surrounding region and potentially able to inhabit particular ecological conditions. This approach has been applied in estimations of populations of ectoparasites based on the environmental or host-associated characteristics of a region, such as air temperature, precipitation, and regional host species richness [97], or influences of host traits, such as degree of sociality, shelter structure, and geographic range size [98].

A method for estimating the flea infrapopulation size on black-tailed prairie dogs (*Cynomys ludovicianus*) was recently conceptualized and may be broadly applicable to other hosts and parasites. The utility of this method increases with decreasing infrapopulation sizes [99].

## 7. Coevolution and Phylogeny

Host specificity and distribution constitute some of the most important evidence for the study of coevolution between fleas and mammals. The correlation between fleas and hosts is so close that, in many instances, it seems that they must have been intimately associated for countless years [100]. Due to their high specificity, lice are the most suitable ectoparasites for

studies of coevolution with hosts, following Fahrenholz's parasitophyletic rule, expressed in [101]: *parasite phylogeny mirrors host phylogeny*. Nevertheless, this author [100] concluded that a significant number of species of Siphonaptera, representing about 14% of the known siphonapteran fauna, co-evolved with their mammalian hosts, and listed 92 genera and 233 species of fleas that are ultraspecific to 122 mammal species. Another 183 species are restricted to a single host genus. The fact that a flea species uses more than one species in the same genus as hosts, following Manter's rule—*parasites evolve more slowly than their hosts* (Figure 5)—might present some epidemiological importance, for example, if one of the hosts becomes a reservoir or serves as an experimental model for the study of a certain disease.

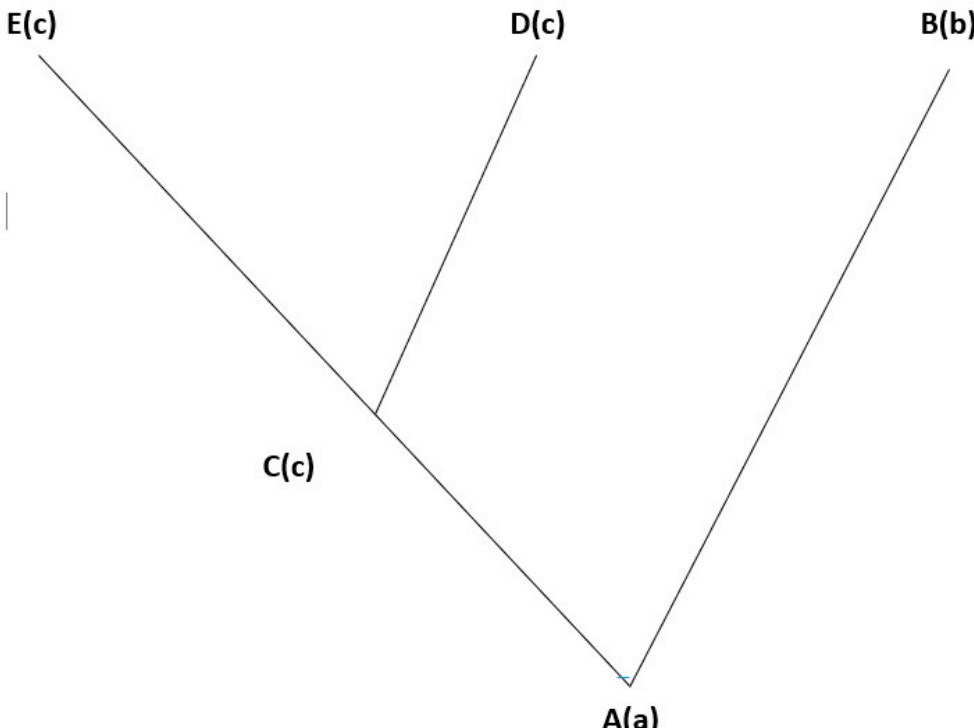

**Figure 5.** Manter's parasitophyletic rule: flea species (small letters); host species (capital letters). Hosts D and E form a monophyletic group, sharing the same flea species c.

In general, phylogenetic studies with fleas are based on morphological and molecular data, despite other approaches that could be developed using ectoparasites as characters and evaluating similarities. Two techniques can be used for the study of similarities [102]: the Q technique, which evaluates similarity between taxa by means of characters, and the R technique, which, conversely, compares characters by means of the studied taxa. The former method was used to evaluate the phylogenetic relationships among 11 species of *Tunga* and infestation on groups of mammalian hosts [103]. Similarly, the hosts were used as characters (R technique) to obtain a cladogram for the *Tunga* species (Figure 6), in which specific flea species would be autapomorphies and fleas parasitizing distant taxonomic groups would be homoplasies.

Although with small distortions, the cladogram clearly shows a clade composed of the species included in the *caecata* group (*caecigena*, *callida*, *bossi*, *caecata*, *libis*, and *monositus*), currently subgenus *Brevidigita* [104], as opposed to the *penetrans* group, subgenus *Tunga*.

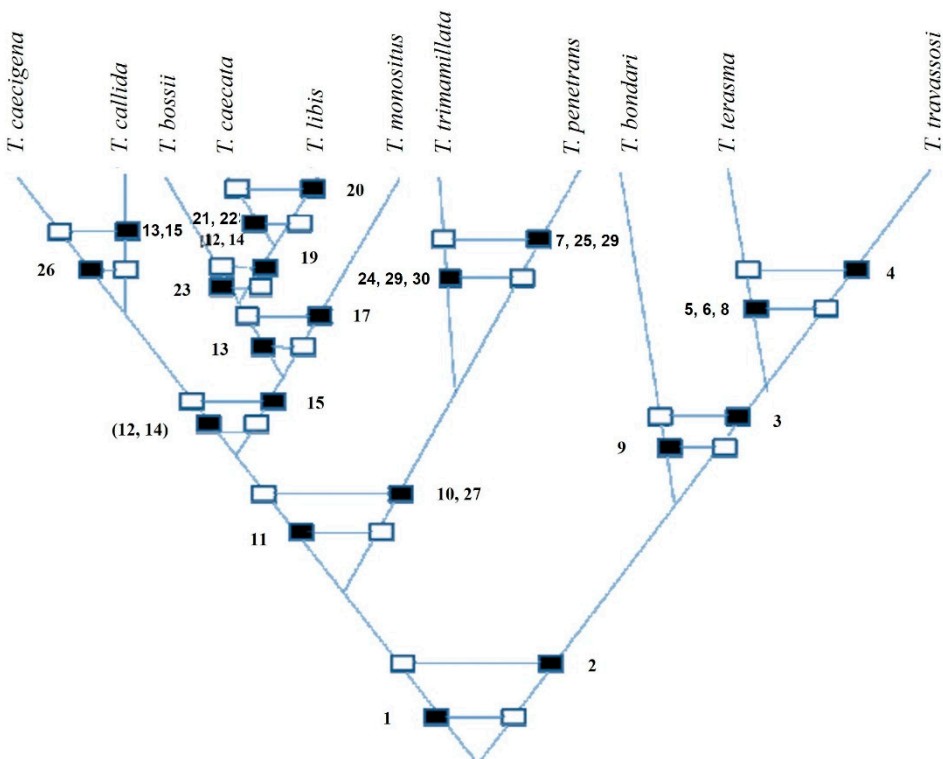

**Figure 6.** Cladogram of some species of *Tunga* based on groups of hosts used as characters (numbers 1 to 30). White bars indicate plesiomorphic states of the character. Black bars indicate apomorphic states. Heterobathmy of characters with synapomorphies and autapomorphies can be seen in this scheme of argumentation. Numbers in parentheses indicate homoplasies. Source: de Avelar [103].

## 8. Literature and Art

Fleas have been immortalized not only in science, but also in literature and the arts, providing entertainment, disseminating culture, and bringing fame to the creators. There is insufficient space here to cite all these productions, so only the main instances will be considered. Such productions occur in several poems and literary works, paintings, films, jokes, and beliefs, which, in a certain way, has contributed to knowledge of the Siphonaptera order and its dissemination [105,106].

In literature, fleas are featured in the fables of Aesop ("*The abbot and the flea*", "*The flea and the ox*", "*The flea and the wrestler*"), Phaedrus ("*The flea and the camel*") and La Fontaine ("*The man and the flea*"). Questions about the rights of fleas were presented in 1768 by Johan Wolfgang von Goethe in "*Juristiche Abhandlung über die Flöhe*". The novel "*The Plague*", originally published in 1947, enshrined Albert Camus [107] as one of the foundational writers of modern literature. The book tells the story of the inhabitants of Oran, Morocco, who were struck by the plague, which decimates the population, and their fight against the transmitting rats and fleas. Several poems addressing fleas have been published, such as "*A Budget of Paradoxes*" by Augustus De Morgan in 19th century; "*The Flea*", by John Dorne (16th century); and "*Fleas*" and "*A Flea in a Fly in Flue*", by Frederic Ogden Nash, an American poet who lived between 1902 and 1971. However, there are doubts about the authorship of "*Fleas*"—considered to be the shortest poem in the world—which can be attributed to Strickland Gillilan (1869–1954).

Among the arts, the main paintings involving fleas and their respective painters, exhibition locations and year of production are presented in Table 5. Two films are of particular relevance to fleas. One, "Limelight", includes a memorable scene of flea tamers "Phyllis" and "Henry", and was directed and produced by Charles Chaplin in 1952, who also starred in the film. The other, "The Death of the Director of the Flea Circus", directed

by Thomas Koerfer in 1973, is about a man who makes a living at a flea circus, until his fleas are poisoned, forcing him to change his activity [108].

**Table 5.** Paintings about fleas.

| Name | Painter | Location | Date |
|---|---|---|---|
| The Triumph of Death | Pieter Brugel | Museo del Prado, Madrid | 1562 |
| The Flea Hunt | Gerrit von Hontherst | Dayton Art Institute, Dayton | 1621 |
| A Pleasant Woman Picking Flea off a Dog | Adrian Brower | The Metropolitan Museum of Art, New York | ca. 1626-27 |
| The Flea Catcher | Georges de La Tour | Musée Lorrain, Nancy | ca. 1630-34 |
| Pleasant Girl Catching a Flea | Giovanni Battista Piazzetta | Museum of Fine Arts, Boston | 1715 |
| Woman Searching for Fleas (Figure 7) | Giuseppe Maria Crespi | Musée Louvre, Paris | ca. 1710-30 |
| The Flea Hunter | Giuseppe Maria Crespi | Galleria degli Uffizi, Florence | 1720 |
| The Ghost of a Flea | William Brake | Tate Gallery, London | ca. 1819-20 |
| A Man Perceived by a Flea | Steven Campbell | National Gallery of Scotland, Edinburgh | 1985 |

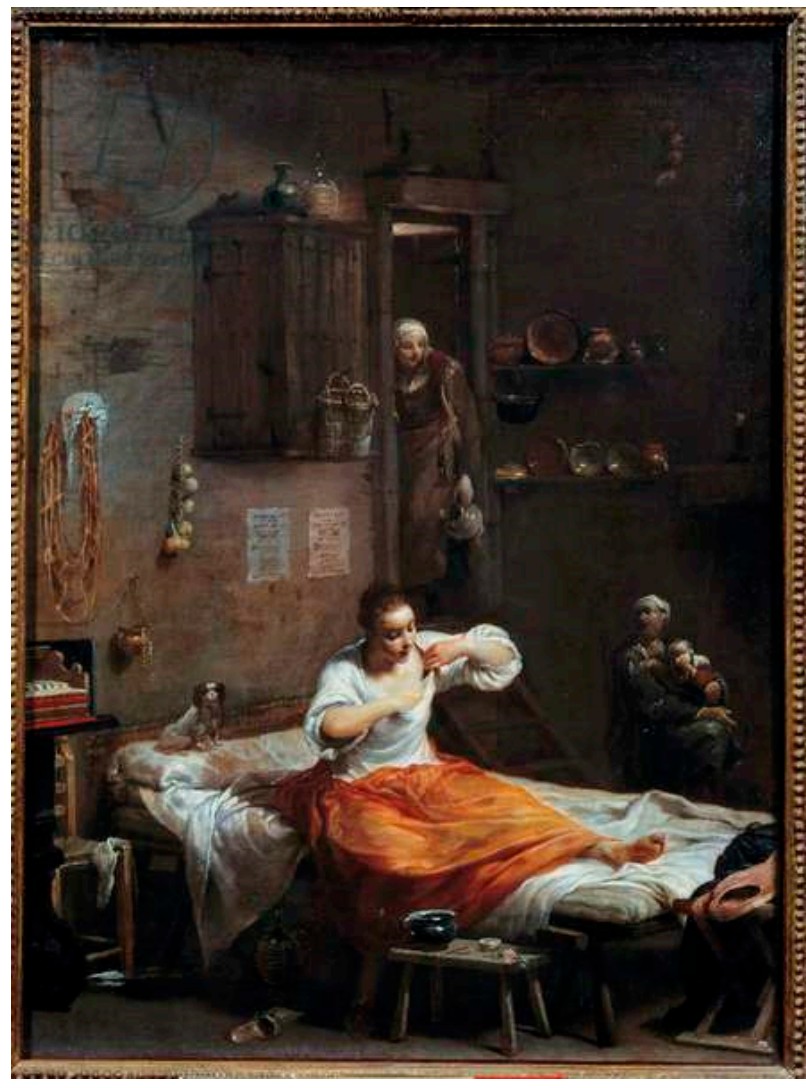

**Figure 7.** Woman searching for fleas by Giuseppe Maria Crespi. Source: web, licensable figure.

Another delicate art, that of dressing fleas in tiny costumes no more than 5 mm in height, also known as "Pulgas Vestidas", began in Mexico over two centuries ago. The fleas were set in matchboxes in scenes as married couples in miniature, with the bride sporting a

long veil and the groom in his best suit. They were widely sold to tourists when visiting Mexico in the early 20th century.

The most fantastic and well-known art related to these insects is the Flea Circus, which was very active in 19th century Germany and England. The show was based on the ability of fleas to, when connected to a tiny carriage by a wire, displace considerable weights. In this show, fleas dance, juggle, and propel a Ferris wheel in attempts to escape.

It is important to note that this means of fun, tying fleas, allowed W. Nöller to experimentally demonstrate the transmission of plague by these insects, as cited by Cunha [109].

## 9. Future Directions

There must still be about 500 flea species yet to be described, slightly more than previously thought [30]. Many species are known through only a few records and some only by only one of the sexes. The likelihood of discovering new species increases when multiple hosts are explored across a wide geographic distribution [110]. On the other hand, efforts should be concentrated in regions with a small number of known species, or that have been little explored, such as Brazil, which is endowed with a rich mastofauna and several biomes, refuge areas, and dispersal centers (Table 4). An example is the genus *Tunga*, of Neotropical origin, which contains fourteen species, of which five have been described in the last 20 years, four of these only between 2012 and 2014 [104]. Four species are known only from their neosomes [111], which are pregnant females hypertrophied in the skin of their hosts (Figure 8) [112].

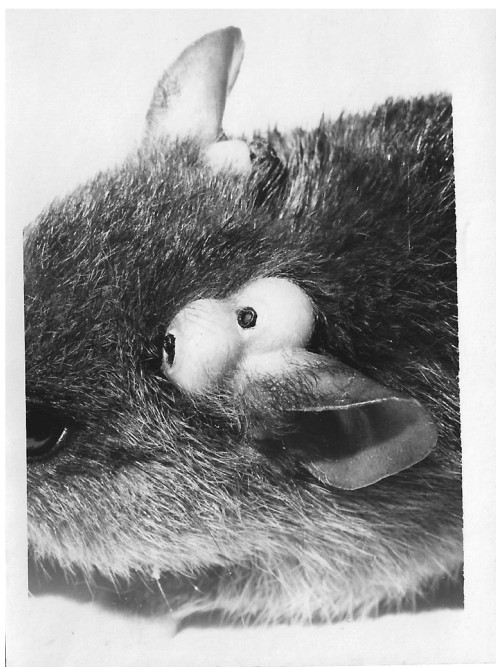

**Figure 8.** Neosomes of *Tunga* (*Brevidigita*) sp. on the ear of *Oligoryzomys nigripes*.

In view of the constant environmental transgressions that occur day by day all over the world, characterized by deforestation, indiscriminate fires, mining, construction of highways and hydroelectric plants, and climate change, there is an urgency for species to be known, before they become extinct [113]. Particular attention should be paid to mammal species hosting specific ectoparasites and to those threatened or at risk of extinction. Ecological parameters, such as richness, abundance, and prevalence of fleas, could assist conservation studies, for example, by temporally comparing a single locality or habitat, thus revealing any changes over time [95]. These ecological data should, whenever possible, be included in taxonomic studies, since statistical methods can categorize infestation types. Knowledge of exclusive, primitive, or primary hosts, together with their respective ectopar-

asites, opens interesting prospects for studying co-association and may reveal the processes of co-evolution (common phylogenies) or co-accommodation (simultaneous ecological adaptations) [100,101].

Since fleas leave their hosts when disturbed or confined for a long time inside traps, the absence of fleas on small mammals captured in the field must be credited to the time elapsed before trap inspection. On the other hand, it can also be an indicator of environmental transgressions, because the biological cycle of fleas develops in the soil, outside the hosts. When fleas are collected, and before being fixed in ethanol, they should be kept for some time in glass or test tubes covered with gauze or polyethylene caps, to allow, in a few hours, for oviposition by pregnant females and consequent post-embryonic development [113]. The fleas can then be transferred to tubes containing 70 or 80% ethanol for future taxonomic study. It should be noted that larvae are known for no more than 70 flea species worldwide. Along the same lines, research to elucidate biological cycles should be undertaken. Except for the species of Pulicidae and *T. penetrans*, which already have well-known data, biological studies should be carried out with the purpose of clarifying how certain nutritional and environmental factors influence the development of such stages. When delineated as a means, such a study may provide new taxonomic insights with regard to (i) descriptions of yet unknown sexes; (ii) descriptions of immature forms; (iii) documentation of variation; and (iv) elucidation of possible reproductive phenomena (e.g., parthenogenesis) [113].

Long ago, fleas were seen only as nefarious insects due to their role in the transmission of the most devastating disease in all human history. Currently, with the plague under control and effective prophylaxis and adequate epidemiological surveillance at hand, flea research, that had been essentially taxonomic and epidemiological, has taken a new course, with incursions into ecology, evolution, phylogeny, and physiology. Consequently, the use of infestations to address biological questions has been growing, as the quest for new knowledge drives new studies for such purposes. To paraphrase Einstein, *imagination is more important than knowledge*!

**Funding:** This research received no external funding.

**Institutional Review Board Statement:** Not applicable.

**Data Availability Statement:** Not applicable.

**Acknowledgments:** The author thank the Springer Nature for the permission to reuse the Table 1, *Fleas and Diseases*, that is not part of the governing OA license, published by P. M. Linardi as a chapter in the book *Arthropod Borne Diseases*, edited by C. B. Marcondes (2017: 517-536). He also thank the Research Trends for the permission to reuse the Figure 4, published by P.M. Linardi and J. R. Botelho in Trends in Entomology (2012, 8:53-62), and to Daniel Moreira de Avelar for the Figure 6, *Tunga* species cladogram.

**Conflicts of Interest:** The authors declare no conflict of interest.

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
