# Peer review of "Fleas as Useful Tools for Science"

_diversity, doi:10.3390/d15111153_

Round 1
Reviewer 1 Report
Comments and Suggestions for Authors
Very high-quality manuscript, excellent review of the literature.
I have only minor comments:
- page 11, row 5. I advice better start the sentence: "Evolutionary primitive hosts..;
- page 17, Figure 7. According to me, the addition of information is missing, e.g. why some of rectangles are black, other white, why are there several numbers in brackets?
- page 18, second row, probably Gillilan (not Gillilen)
Reviewer 2 Report
Comments and Suggestions for Authors
The present manuscript is an excellent review of the current knowledge of fleas, but it also provides an interesting final reflection about the need to increase and strengthen studies on these ectoparasites.
I consider that it can be published in Diversity without modifications.
Author Response
Thanks for your comments

Reviewer 3 Report
Comments and Suggestions for Authors
Comments to the manuscript diversity-2709388 intended as a review in Diversity entitled ‘Fleas as useful tools for science’ by PM Linardi. The author presents an interesting manuscript reviewing mammal fleas and on how fleas can be used in applied research. It appears as unfinished business that the focus in (almost) only on mammal fleas. The review would be strengthened a lot by mentioning all fleas.
A few comments and suggestions:
Abstract: The statement ‘Adult fleas are blood-feeding insects that essentially infest mammals’ is not correct it should be ‘Adult fleas are blood-feeding insects that essentially infest mammals and birds’.
Additionally, it should be stated that this review only includes fleas on mammals. But on page 7 there is a three lines mentioning bird fleas!
Table 1: Please add references as an additional column.
Reference 7 cannot be found by searching litteratur databases; please add another reference and reconsider the statements linked to the reference. The claimed use by US in Korean war has not been confirmed and should be deleted.
Page 10: ‘Parasitism on birds is secondary and must have originated from fleas parasitizing mammals.’ Please add reference. Questionable statement, but probably correct statement. Please add some evolutionary arguments for this statement. Maybe list number of species and level of diversity in species between mammal and bird fleas.
Page 11: ‘primitive hosts’ and ‘primitive fleas’ does not appear scientifical correct classification. Please change to a wording based on evolutionary principles
Page 11: ‘Approximately 600 flea species…’. Does this only cover mammals?
Figure 4. Why is Figure 4 included. Purpose? Please delete, it is not adding to the manuscript.
Figure 15: ‘countless eons’. Please change to years, which would be scientifically more correct.
